# Antibacterial Activity of Trypsin-Hydrolyzed Camel and Cow Whey and Their Fractions

**DOI:** 10.3390/ani10020337

**Published:** 2020-02-20

**Authors:** Ruixue Wang, Zhihao Han, Rimutu Ji, Yuchen Xiao, Rendalai Si, Fucheng Guo, Jing He, Le Hai, Liang Ming, Li Yi

**Affiliations:** 1Key Laboratory of Dairy Biotechnology and Bioengineering, Ministry of Education, College of Food Science and Engineering, Inner Mongolia Agricultural University, Hohhot 010018, Inner Mongolia, China; 15754888509@163.com (R.W.); hanzhihao1749@163.com (Z.H.); yeluotuo1999@vip.163.com (R.J.); xycimau@163.com (Y.X.); sirendalai_imau@163.com (R.S.); guofucheng1101@163.com (F.G.); hejing1409@163.com (J.H.); haileaijia@163.com (L.H.); bmlimau@163.com (L.M.); 2Camel Research Institute of Inner Mongolia, Alxa 737300, Inner Mongolia, China

**Keywords:** camel milk, cow milk, ultra-filtration, chromatography, amino acid, antimicrobial peptide

## Abstract

**Simple Summary:**

Camels are an important part of the ecosystem in desert areas. Camels can survive well under difficult conditions, therefore, they play a key role in local herdsmen’s production, life, and economic structure. China’s Inner Mongolia region has unique environmental and geographical advantages, making it suitable for camel breeding. Camel milk has high nutritional value and unique functional characteristics. It not only has low sensitization, but also contains various immune active factors with high medicinal value. Currently, there are various products derived from cow and goat milk, but few related to camel milk, providing an opportunity for development. This study used trypsin to hydrolyze the whey proteins of camel milk, separated and purified peptide fragments with antibacterial activity, and conducted a comparative study with cow milk. The present study provided new ideas for the use and development of camel whey protein hydrolysates and their dextran purification fractions, and indicated the future development of these peptides as nutritional additives or food preservatives.

**Abstract:**

Antibacterial peptides were isolated and purified from whey proteins of camel milk (CaW) and cow milk (CoW) and their antimicrobial activities were studied. The whey proteins were hydrolyzed using trypsin, and the degree of hydrolysis was identified by gel electrophoresis. The whey hydrolysate (WH) was purified using ultrafiltration and Dextran gel chromatography to obtain small peptides with antibacterial activity. The effect of the antimicrobial peptides on the morphology of bacterial strains was investigated using transmission electron microscopy. Their amino acid composition and antimicrobial activities were then determined. Polypeptides CaWH-III (<3 kDa) and CoWH-III (<3 kDa) had the strongest antibacterial activity. Both Fr.A2 (CaWH-Ⅲ’s fraction 2) and Fr.B1 (CoWH-Ⅲ’s fraction 1) had antibacterial effects toward *Escherichia coli* and *Staphylococcus aureus*, with minimum antimicrobial mass concentrations of 65 mg/mL and 130 mg/mL for Fr.A2, and 130 mg/mL and 130 mg/mL for Fr.B1, respectively. The highly active antimicrobial peptides had high amounts of alkaline amino acids (28.13% in camel milk Fr.A2 and 25.07% in the cow milk Fr.B1) and hydrophobic amino acids. (51.29% in camel milk Fr.A2 and 57.69% in the cow milk Fr.B1). This results showed that hydrolysis of CaW and CoW using trypsin produced a variety of effective antimicrobial peptides against selected pathogens, and the antibacterial activity of camel milk whey was slightly higher than that of cow milk whey.

## 1. Introduction

The main factor affecting food shelf life is microbial contamination, which might also affect the health of consumers. Therefore, preservatives containing chemicals are usually used to delay microbial growth or avoid the spoilage caused by chemical changes. In recent years, given the aim to reduce the use of chemical agents, consumers have expressed more interest in foods containing natural preservatives. Therefore, the use of natural antiseptics instead of chemical preservatives has become a hot topic [1].

Numerous authors have reported that many different types of peptides can be released from milk whey proteins by proteolytic enzyme digestion [2]. Antibacterial peptides are bioactive peptides obtained from the enzymatic hydrolysis of different animal-derived milk proteins. For example, one study showed that human gastrointestinal enzymes produce different peptides from goat whey than do non-human enzymes, and that the hydrolysates have higher antibacterial activities than the pure peptides. The antibacterial activity against pathogens and probiotics suggests that whey might have host-protective activity [3]. The bacteriostatic effect of the degradation of goat whey protein by human protein hydrolase on *Listeria monocytogenes* has been studied. Undigested goat whey did not show a significant inhibitory effect; however, the antibacterial effect of goat milk hydrolysate was strongest in the duodenum during digestion [4]. The peptide lactoferrin B, produced by pepsin digestion of bovine lactoferrin (LF), inhibits and inactivates both gram-negative and gram-positive bacteria, such as *Klebsiella pneumonia*, *Proteus vulgaris*, *Escherichia coli*, and *Salmonella enteritidis* [5]. Bovine whey beta lactoglobulin (β-Lg) has been hydrolyzed by trypsin to produce four bactericidal peptides [6]. Human and bovine LF hydrolysates effectively inhibit many pathogens [7].

Camel milk whey protein is regarded as a good source of bioactive peptides. It contains α-lactalbumin (α-LA; 2–7 g/L; 14.42 kDa), serum albumin (SA; 0.40 g/L; 69.6 kDa), LF (0.02–2.1 g/L; 75.25 kDa) and lysozyme (0.00015–0.005 g/L), among many other protective proteins [8]. These proteins are highly stable, possess antimicrobial activity, and are present in larger amounts in camel milk compared with that in cow milk [α-LA (1.1 g/L), SA (0.36 g/L), LF (0.17 g/L), lysozyme (0.00007 g/L), and lactoperoxidase (LP) (0.03 g/L)] [9]. In addition, El-Agamy et al. [10] detected more of these protective proteins in camel milk whey than in sheep, goat, or buffalo milk. Higher activity of LF against *E. coli O157:H7* was detected in camel milk compared with the LF isolated from other milk sources [11]. α-Chymotrypsin and alcalase hydrolyze camel milk to yield hydrolysates with high antimicrobial activity [12]. The main mechanism underlying this process may be the release of bioactive peptides and antibacterial fragments, or the high resistance of antibacterial proteins to pepsin and trypsin [2]. Enzymatic hydrolysis of camel milk improved its antibacterial and antioxidant activities [12]. The antimicrobial characteristics of camel milk whey protein hydrolysate have been evaluated [12,13,14], further demonstrating that camel milk is a good source of antibacterial proteins.

In this study, we aimed to investigate the antibacterial activities of cow and camel whey protein trypsin hydrolysates (CoWH and CaWH, respectively) against selected spoilage and pathogenic bacteria, including *Salmonella typhimurium*, *Staphylococcus aureus*, *Escherichia coli*, and *Streptococcus mutans*.

## 2. Materials and Methods

### 2.1. Materials

Milk from cows (3.62% fat, 3.54% total protein) and camels (4.23% fat, 3.47% total protein) was obtained from the pasturing area of Alxa City, Inner Mongolia Governorate, China. The fat and whole protein in cow and camel milk were analyzed by using milk composition analyzer (MilkoScan FT1, FOSS, Beijing, China) [15]. The milks were heated to 40 °C and the fat was skimmed off after centrifugation at 10,000 rpm/min, for 20 min, at 4 °C. 1 mol/L HCl was added to pH 4.6 and then precipitated by centrifugation at 8000 rpm/min, at 4 °C for 20 min. The whey was then collected, dialyzed against phosphate buffer (50 mM, pH 7.8). The samples were pre-frozen at 80 °C for 24 h and then placed in a freeze-dryer (SCIENTZ-10N, Guangdong, China) and vacuum freeze-dried at 50 °C and 2–10 Pa to obtain freeze-dried powder samples, and stored at −20 °C until use. Porcine trypsin (250 U/g) was obtained from Solarbio (Beijing, China). All other chemicals were of analytical grade.

### 2.2. Camel Whey (CaW) and Cow Whey (CoW) Enzymatic Hydrolysis

Lyophilized CaW and CoW were dissolved in 0.01 M Na_2_HPO_4_-NaH_2_PO_4_ buffer (pH 8.0) and trypsin batch hydrolysis (enzyme /substrate (E/S) ratio = 1:100, *w*/*w*) was performed at 40 °C and pH 8.0 ± 0.1 for 240 min. A previously published method [16] was used to determine the degree of hydrolysis (DH) after 30, 60, 120, 180, and 240 min. At the end of the hydrolysis, the enzyme is inactivated by heating in a boiling water bath for 10 min. The hydrolysate was subjected to centrifugation (5000 rpm/min, 30 min, 4 °C), and the insoluble substrate fragments were removed. The supernatant was subjected to freeze-drying and stored at −20 °C for further use. CaW and CoW were hydrolyzed in triplicate. The DH was determined using the *o*-phthaldialdehyde (OPA) method described by Jrad et al. [2].

### 2.3. Sodium Dodecyl Sulphate-Polyacrylamide Gel Electrophoresis

The method of Laemmli [17] was used to carry out sodium dodecyl sulphate-polyacrylamide gel electrophoresis (SDS-PAGE) under reducing conditions. CaW and CoW hydrolysates (20 mg) were mixed with 500 mL of reducing sample buffer (0.3 M Tris-HCl (pH 6.8), 50% glycerol, 5% SDS, and 100 mM dithiothreitol), heated for 5 min at 95 °C. Ten microliters of the denatured proteins were loaded into prefabricated 12% acrylamide gels and electrophoresed for 30 min at a constant 90 V and then for 60 min at 120 V. The gel was fixed (overnight at room temperature), the proteins were stained using Coomassie brilliant blue R250 (overnight at room temperature), and then the gel was decolorized [18].

### 2.4. Fractionation of CaW and CoW Hydrolysate

#### 2.4.1. Ultrafiltration

CaWH and CoWH were subjected to ultrafiltration fractionation at 4 °C (UFC910096, Millipore, Shanghai, China) with molecular weight cutoff (MWCO) membranes of 3 and 10 kDa (Millipore, Hangzhou, China). The collected peptide fractions (CaWH: CaWH-I > 10 kDa, CaWH-II 3–10 kDa, and CaWH-III < 3 kDa; CoW: CoWH-I > 10 kDa, CoWH-II 3–10 kDa, and CoWH-III < 3 kDa) were lyophilized and then subjected to dextran gel filtration chromatography.

#### 2.4.2. Dextran Gel Filtration Chromatography

CaWH and CoWH were subjected to dextran gel filtration chromatography (DGFC) using a Sephadex G-25 Fine, cross-linked dextran (26–100 mm, HiPrep 26/10 Desalting) using a fast protein liquid chromatography (FPLC) system (AKTA purifier, Uppsala, Sweden). Samples comprising CaWH and CoWH at 200 mg/mL in distilled water were injected at a flow rate of 5 mL/min and detect at 280 nm. Fractions (25 mL) were collected, subjected to freeze-drying, and then their antibacterial activities were assessed. a bicinchoninic acid protein analysis kit (Beyotime, Shanghai, China) was used to determine the protein concentrations.

### 2.5. Antibacterial Activity

#### 2.5.1. Bacterial Strains Used

To assess the antibacterial activity of CaW, CoW, CaWH, CoWH, and the Sephadex G-25 gel filtration column fractions, we used two gram-negative bacteria, *Escherichia coli* (ATCC 25922) and *Salmonella typhimurium* (ATCC 50115); and two gram-positive bacteria, i.e., *Streptococcus mutans* (ATCC 25175) and *Staphylococcus aureus* (ATCC 25923) The Guangdong Culture Collection Center (Guangdong, China) provided these bacterial strains.

#### 2.5.2. Disc Diffusion Assay

The method of Bauer and et al. [19] was used to assess the antimicrobial activity. Samples of CaW and CoW, CaWH, and CoWH (200 mg/mL in distilled water) were filtered through 0.22-μm cellulose acetate membrane filters (Solarbio, Beijing, China). *E. coli*, *S. aureus*, *Salmonella typhimurium* and *Streptococcus mutans*) were grown at their optimum temperature (37 °C) overnight in Brain-Heart Infusion broth medium. Thereafter, the bacteria were diluted to about 10^6^ colony forming units (cfu)/mL and coated on the selective agar medium. The peptide samples (150 µL) were loaded onto sterile papers (5 mm diameter) and placed on the agar surface. After storage for 24 h at 4 °C the plates were incubated at the optimal temperature for each strain for 12 h. The diameter of the bacterial inhibition zone indicated the antibacterial activity.

#### 2.5.3. Determining the Minimum Inhibitory Concentration

A previously described method [20] was used to determine the minimum inhibitory concentration (MIC) of selected samples. Each sample was twice decreasing concentration dilution, and each of the 150 µL of the bacteriostatic solution was applied to the inoculated agar as described in Section 2.5.2. The lowest concentration (maximum dilution) of the test sample that showed a visible clear area on the selective agar medium was considered to be the MIC.

### 2.6. Transmission Electron Microscopy

#### 2.6.1. Preparation and Fixation of Bacteriostatic Solution

The protein concentration of each antimicrobial peptide component was adjusted to 200 mg/mL in NB broth medium, and the configured solution was filtered and sterilized by 0.22 μm aseptic filter. The 3 mL of each configured antimicrobial peptide solution is respectively added to the bacterial solution of the same volume, so that the final concentration of each component is 3 mg/mL, and the final concentration of the bacterial solution is 10^5^ cfu/mL. The bacteria were collected by centrifugation after being cultured in a constant temperature incubator at 37 °C for 8 h, and washed three times with 0.1 M PBS. Finally, 2.5% glutaraldehyde was added and fixed for 24 h.

#### 2.6.2. Embedding of Bacteriostatic Solution

Each bacteriostatic fixing solution was washed with 0.1M PBS for 3 times, fixed with 1% Osmic acid at 4 °C for 3 h, and then washed again for 3 times. Then, step by step dehydration was carried out with ethanol, propylene oxide was replaced. Spurr resin was impregnated and encapsulated, and polymerized in an oven at 70 °C.

#### 2.6.3. Section Staining Observation

The embedded blocks of different bacteriostatic solutions were sliced with an ultra-thin slicer with a thickness of 70 nm, then stained with uranyl acetate and lead citrate, and the samples were observed and photographed under transmission electron microscope [21].

### 2.7. Amino Acid Composition Determination

The method of Siswoyo et al. [22], with slight modifications, was used to determine the amino acid composition. The lyophilized peptide was dissolved in distilled water at 1 mg/mL. Fifty milliliters of this solution was dried under a vacuum and then hydrolyzed in 6 M HCl with 0.1% phenol for 24 h at 110 °C. The internal standard used was Norleucine (Sigma Aldrich, Inc., St. Louis, MO, USA). Upon completion of hydrolysis, the samples were vacuum dried, resuspended in application buffer, and applied to an automatic amino acid analyzer (HITACHI 835-50, Tokyo, Japan). Triplicate measurements were made and the data were averaged. In all cases, the standard deviation was less than 2%. The original data were manually integrated by the instrument’s own LabSolution software. First, an amino acid mixed standard was prepared, and the corresponding external standard method was established. The method was then used to automatically integrate the chromatogram of the test sample, and the molar percentage of the amino acid composition of the sample was obtained. The significance analysis of the data was carried out by using IBM-SPSS statistics v.23 software (Senchtech, Beijing, China).

## 3. Results and Discussion

### 3.1. Preparation of CaWH and CoWH

The DH value and SDS-PAGE were used to estimate the extent of trypsin degradation of the proteins. The molecular weights of the proteins were estimated by comparing CaW and CoW proteins with standard labeled proteins having molecular weights of 180–14.4 kDa. Electrophoretic profiling of CaW revealed three components corresponding to LF, α-LA, and SA (Figure 1, lane 1). Salami et al. [23], Saliha et al. [24], and Tagliazucchi et al. [25] observed similar results. The estimated molecular mass values of LF, SA, α-LA, and β-LG in CoW are shown in Figure 2 (lane 1). β-LG was absent from camel milk, and the predominant protein in cow milk was whey protein [26], which was consistent with the results reported by El-Hatm [27].

The CaWH obtained after 180 min of degradation had a higher DH (22.88%) compared with those obtained after 30, 60 and 120 min of degradation (8.41%, 9.82% and 10.23%, respectively) (Figure 3). The CoWH obtained after 240 min of degradation had a higher DH (20.93%) compared with those obtained after 30, 60, 120 and 180 min of degradation (7.89%, 8.36%, 9.44% and 10.63%, respectively) (Figure 3). For camel milk, digestion with trypsin for 180 min completely hydrolyzed the whey protein fractions (Figure 1, lanes 2–4), while those of cow milk were hydrolyzed within 240 min (Figure 2, lanes 2–5). These results indicate that CaW is easier for trypsin to hydrolyze than CoW; in addition, CaW is hypoallergenic because it lacks β-LG.

### 3.2. Ultrafiltration Fractionation (UF) of CaWH and CoWH

The membrane filtration process UF uses pressure or a concentration gradient to force substances through a semi-permeable membrane. UF is widely used to obtain the target fraction of a protein hydrolysate using a molecular weight cut-off (MWCO) membrane [28]. In this study, >10 kDa CaWH-I and CoWH-I, 3–10 kDa CaWH-II and CoWH-II, and <3 kDa CaWH-III and CoWH-III were fractionated from CaWH and CoWH using 10 kDa and 3 kDa MWCO membranes. CaWH, CoWH, and their six collected fractions were lyophilized and subjected to antibacterial activity assessment (Table 1).

After UF, compared with the other fractions, CaWH-III and CoWH-III demonstrated significantly higher antibacterial activity (*p* < 0.05). Compared with that against the other bacterial strains, the inhibitory effect was significantly higher against *E. coli* (ATCC 25922). Camel milk displayed a higher antibacterial effect compared with that of CaWH-III and CoWH-III; however, the results did reach statistical significance. This result is similar to that of Jrad [29]. Many studies have shown that hydrolysates with high biological activity comprise low molecular weight peptides [30,31]. The results of this experiment are consistent with previous reports; i.e., low molecular weight protein hydrolysates and peptides have higher antibacterial activity, and interact more effectively with free radicals, thus enhancing the antibacterial process [32,33].

### 3.3. CaWH-III and CoWH-III Gel Filtration Chromatography

Substances with different molecular weights are commonly separated using gel filtration, particularly to separate peptides from protein hydrolysates [34]. In the present study, a Sephadex G-25 column was further used to purify CaWH-III and CoWH-III (Figure 4). Subsequently, CaWH-III was separated into two subfractions termed Fr. A1 and Fr. A2 (Figure 4A,B, respectively): Fr. A2 showed significantly higher antimicrobial activity than Fr. A1 (*p* < 0.05) (Table 2). CoWH-III comprised only one fraction (Fr. B1). Strong antibacterial activity against E. coli was observed for Fr. A1, Fr. A2, and Fr. B1, with the activity of Fr. A2 being higher than that of Fr. B1 (Table 2). This was similar to the results reported by Salami et al. [10], who demonstrated that the strongest antibacterial activity against *E. coli* was demonstrated by permeates of camel whey protein hydrolysates from UF (3 kDa rejection). The presence of low mass peptides might explain the strong antibacterial activity of Fr. A2 and Fr. B1. Fr. A2 and Fr. B1 were eluted last; therefore, they most likely contain relatively low mass peptides. In general, most antimicrobial peptides are short (2–20 amino acids). Low mass peptides tend to have higher antibacterial activity than high mass peptides [35].

### 3.4. Minimum Inhibitory Concentrations (MICs) of CaWH-III-Fr.A2 and CoWH-III-Fr.B1

The antibacterial activities of CaW, CoW, CaWH-III-Fr. A2, and CoWH-III-Fr. B1 against two gram-negative bacteria (*E. coli* and *Salmonella typhimurium*) and two gram-positive bacteria (*S. aureus* and *Streptococcus mutans*) were assayed to determine the MIC of the above peptide fractions, hydrolysates, and milks (Table 3). Fr. A2 and Fr. B1 showed potent antibacterial effects with lower MIC values than those of CaW and CoW. The most significantly affected bacteria were *S. aureus* and *E. coli*, with Fr. A2 having the lowest MIC values of 130 and 65 mg/mL and Fr. B1 having MICs of 130 and 130 mg/mL, respectively (Table 3).

In addition, the MIC value obtained for Fr. A2 was significantly lower than that for Fr. B1. The antibacterial effects of peptides produced by different milk sources differed, as did the MIC values. These results showed that camel milk has a similar, or slightly higher, bacteriostatic effect to cow milk; thus, it warrants further research and development.

### 3.5. Transmission Electron Microscopy

The changes to the ultrastructure of bacteria incubated with the antimicrobial peptides was observed using transmission electron microscopy (TEM). Bacterial cell membranes are responsible for many basic functions, such as respiration, transport, osmotic regulation, lipid synthesis, and biosynthesis and crosslinking of peptidoglycans. Membrane integrity is vital for all of these functions, and direct or indirect disruption of the membrane could lead to the formation of pores, metabolic dysfunction, and even cell death [36]. At the lethal concentration of the antimicrobial peptides, observing the integrity of the bacterial membrane using TEM would help to reveal the detailed mechanisms of cell death; however, only a few antimicrobial peptides have been studied ultrastructurally to date [37].

The antibacterial test showed that isolated and purified CaW-Fr. A2 and CoW-Fr. B1 had strong inhibitory activities against *E. coli* (ATCC 25922) and *S. aureus* (ATCC 25923); therefore, these two strains were selected for TEM analysis. The ultrastructure of strains incubated in a 65 mg/mL antibacterial solution containing 10^5^ CFU/mL of *S. aureus* for 9 h, and an *E. coli* culture for 12 h, was observed under TEM. The effect of the antibacterial solution on the bacterial structure differed: CaW-Fr. A2 interacted with *S. aureus* (Figure 5a), such that part of the bacterial body wall was separated, and invagination was severe (Figure 5b). Meanwhile, some of the cells treated with CoW-Fr. B1 were vacuolated; however, the cell membrane was intact, the morphology was largely unchanged, and there was no obvious damage to the bacterial cell membrane (Figure 5c). Following the interaction between CaW-Fr. A2 and *E. coli* (Figure 5d), some bacterial cells appeared to be severely deformed and showed slight separation of their walls (Figure 5e), while only mild plasmolysis was observed in some bacterial cells treated with CoW-Fr. B1 (Figure 5f).

### 3.6. Fr.A2 and Fr.B1 Amino Acid Compositions

In general, substances with higher bacteriostatic activity contain higher levels of hydrophobic and basic amino acids (i.e., positively charged amino acids) [38], because the components of the bacterial cell wall are mostly negatively charged. Substances bind to the cell walls of bacteria through electrostatic interactions, resulting in destruction of the cell wall structure. Hydrophobic amino acids also cleave the bacterial cell wall, resulting in bacterial cell apoptosis [39,40,41]. Therefore, the amino acid composition of a food protein hydrolysate significantly affects its various physiological activities, including antibacterial activities.

Table 4 shows the amino acid profiles of Fr. B1 and Fr. A2. In Fr. A2, the proportion of hydrophobic amino acids was 51.29% and that of basic amino acids was 28.13%. The corresponding values in Fr. B1 were 57.69% and 25.07%, respectively. The results showed that the amino acid compositions of the trypsin hydrolysates were similar in cow and camel milk.

Our results showed that camel and cow milk have similar antibacterial activities and amino acid profiles. In addition, El Hatm [27] reported that the whey protein compositions of camel milk and human milk are very similar. It has been proposed that camel milk is the most suitable milk substitute for infant formulas [27].

## 4. Conclusions

In this study, camel and cow milk whey protein were hydrolyzed using trypsin, and hydrolyzed components with strong antibacterial activity were obtained using separation and purification. The antibacterial effect of camel milk bacteriostatic peptides was slightly stronger than that of cow milk peptides. SDS-PAGE revealed that camel milk does not contain the sensitizing factor β-Lg, making its hypoallergenic properties. Furthermore, camel milk is considered more suitable as a milk substitute for infant formulas. Therefore, future research should focus on purifying single camel whey protein components to evaluate their immune activity. The results of the present study showed that camel whey can be used as a source of natural protein to produce hydrolysates with antibacterial activity. These results will encourage the future development of camel whey hydrolysates, and large-and small-exclusion chromatography derivatives, as nutritional additives or components of antimicrobial drugs. The results also revealed the potential of camel whey and its hydrolysates as antimicrobial agents for human consumption, and as ingredients in health foods to improve their functionality and shelf life.

## Figures and Tables

**Figure 1 animals-10-00337-f001:**
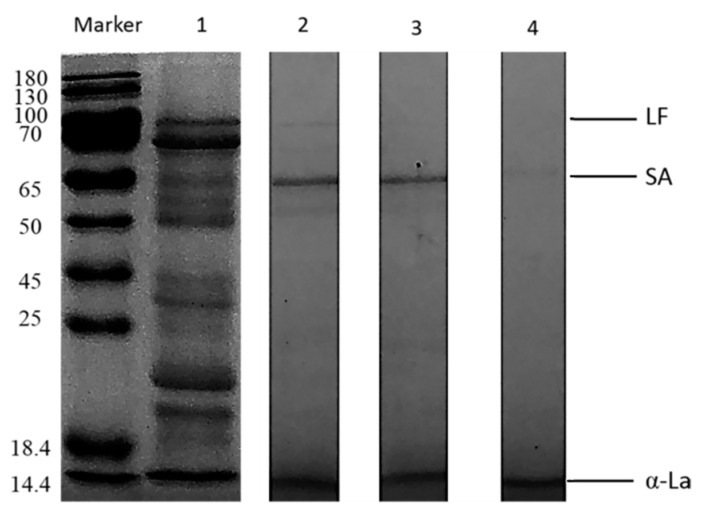
SDS-PAGE (Sodium dodecyl sulfate polyacrylamide gel electrophoresis) of camel whey hydrolysates produced using trypsin for different times: M, protein marker; lane 1, unhydrolyzed camel whey; lanes 2, 3 and 4, camel whey after 60, 120, and 180 min of hydrolysis, respectively. LF, lactoferrin; SA, serum albumin; α-La, α-lactalbumin. Data are representative of three independent experiments.

**Figure 2 animals-10-00337-f002:**
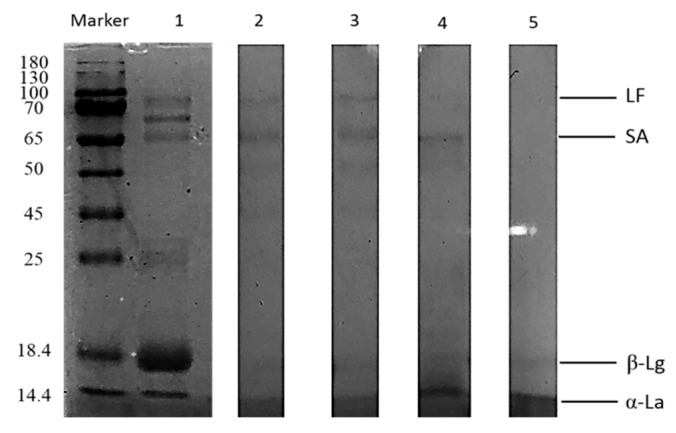
SDS-PAGE (Sodium dodecyl sulfate polyacrylamide gel electrophoresis) of cow whey hydrolysates produced using trypsin for different times: M, protein marker; lane 1, unhydrolyzed cow whey; lanes 2, 3, 4 and 5, cow whey after 60, 120, 180, and 240 min of hydrolysis, respectively. LF, lactoferrin; SA, serum albumin; β-Lg, β-lactoglobulin; α-La, α-lactalbumin. Data are representative of three independent experiments.

**Figure 3 animals-10-00337-f003:**
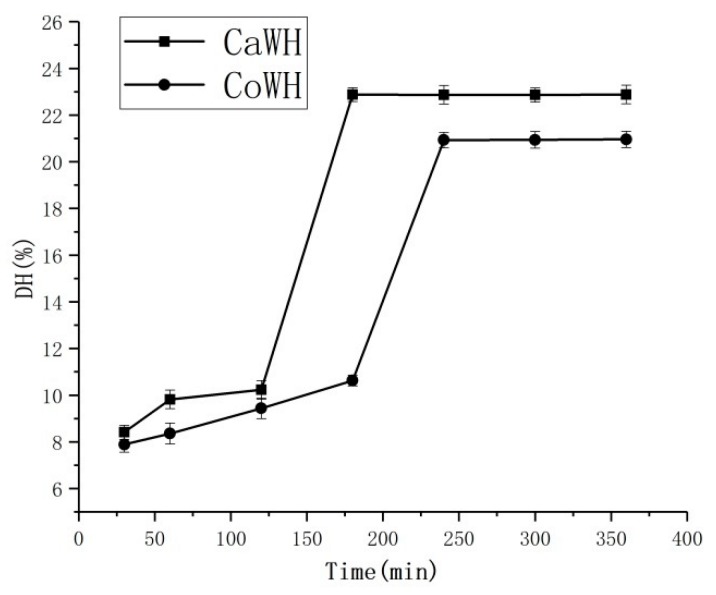
Effect of camel and cow milk whey hydrolysis (CaWH and CoWH) time (30, 60, 90, 120, 180, 240, 300 and 360 min, respectively) on DH (degree of hydrolysis).

**Figure 4 animals-10-00337-f004:**
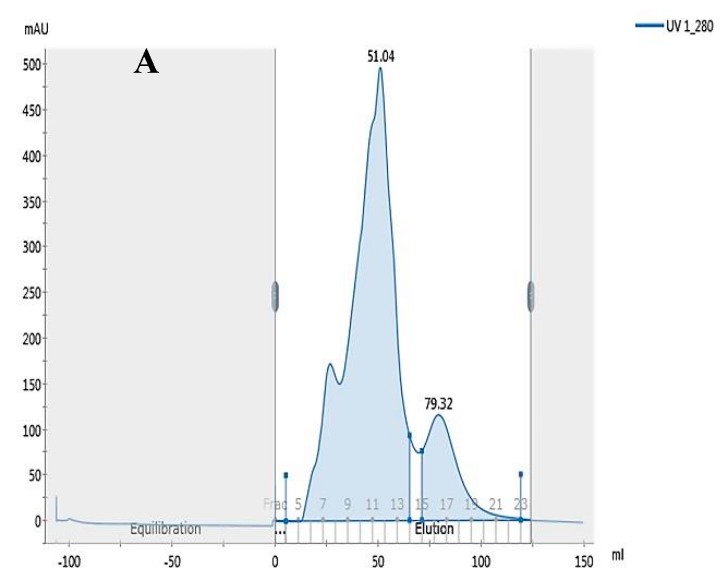
Elution profiles of camel milk whey hydrolysate peptide III (CaWH-III) (**A**) and cow milk whey hydrolysate peptide III (CoWH-III) (**B**) from a Sephadex G-25 chromatography column using Dextran gel.

**Figure 5 animals-10-00337-f005:**
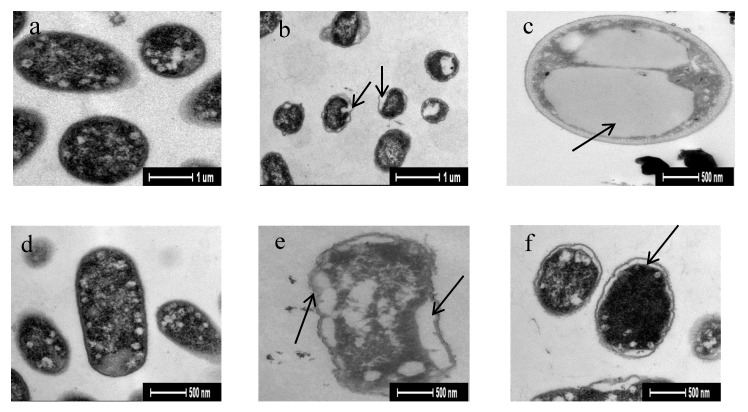
(**a**–**c**): Transmission electron microscopy (TEM) images of the interaction between of Staphylococcus aureus and the combined antibacterial liquid; (**d**–**f**): TEM images of the interaction between of Escherichia coli and the combined antibacterial liquid. TEM analysis was performed on the third replicate of this work.

**Table 1 animals-10-00337-t001:** The relationship between the antibacterial activity of the whey fractions and antibacterial activity in different molecular mass sections.

Strains	Inhibition Zone Diameter (mm)							
	CaWH				CoWH			
	CaWH	CaWH-Ⅰ	CaWH-Ⅱ	CaWH-Ⅲ	CoWH	CoWH-Ⅰ	CoWH-Ⅱ	CoWH-Ⅲ
A	11.90 ^b^ ± 0.30	4.03 ^a^ ± 0.10	10.60 ^b^ ± 0.40	16.30 ^c^ ± 0.20	10.32 ^b^ ± 0.30	3.21 ^a^ ± 0.10	13.60 ^b^ ± 0.20	15.90 ^c^ ± 0.40
B	14.53 ^b^ ± 0.40	6.90 ^a^ ± 0.20	13.99 ^b^ ± 0.30	19.20 ^c^ ± 0.40	13.15 ^b^ ± 0.30	5.30 ^a^ ± 0.20	11.33 ^b^ ± 0.30	18.63 ^c^ ± 0.40
C	8.36 ^a^ ± 0.30	NI	7.81 ^a^ ± 0.20	14.95 ^b^ ± 0.40	8.60 ^a^ ± 0.30	NI	9.01 ^a^ ± 0.10	12.30 ^b^ ± 0.20
D	5.60 ^a^ ± 0.20	NI	5.90 ^a^ ± 0.30	8.30 ^b^ ± 0.50	4.99 ^a^ ± 0.20	NI	4.30 ^a^ ± 0.20	9.60 ^b^ ± 0.30

A: *S.aureus* (ATCC 25923); B: *E.coli* (ATCC 25922); C: *Salmonella typhimurium* (ATCC 50115); D: *Streptococcus mutans* (ATCC 25175); CaWH: camel milk whey hydrolysate; CoWH: cow milk whey hydrolysate. ^a,b,c,^ means with different superscript letters are statistically different in the same line (*p* < 0.05); NI: no inhibition. Note: the value regard as mean ± standard deviation or standard error.

**Table 2 animals-10-00337-t002:** Comparison of the antibacterial activity of cow milk (CoW) and camel milk (CaW) whey protein components.

Strains	Inhibition Zone Diameter (mm)			
	CaWH-Ⅲ		CoWH-Ⅲ	Sterile Water
	Fr.A1	Fr.A2	Fr.B1	
A	17.34 ± 0.40	19.78 ± 0.30	18.35 ± 0.60	NI
B	22.63 ± 0.30	26.90 ± 0.30	24.53 ± 0.40	NI
C	12.80 ± 0.10	14.51 ± 0.20	11.92 ± 0.30	NI
D	9.96 ± 0.20	10.60 ± 0.40	11.13 ± 0.30	NI

A: *S. aureus* (ATCC 25923); B: *E. coli* (ATCC 25922); C: *Salmonella typhimurium* (ATCC 50115); D: *Streptococcus mutans* (ATCC 25175) NI: no inhibition. Note: the value regard as mean ± standard deviation or standard error.

**Table 3 animals-10-00337-t003:** The minimum inhibitory concentration (MIC) of antibacterial peptide extracts from cow milk (CoW) and camel milk (CaW) whey protein.

Strains	MIC (mg/mL)			
	CaW	Fr.A2	CoW	Fr.B1
*S.aureus* (ATCC 25923)	260	130	260	130
*E.coli* (ATCC 25922)	130	65	260	130
*Salmonella typhimurium* (ATCC 50115)	NI	260	NI	260
*Streptococcus mutans* (ATCC 25175)	NI	260	NI	260

**Table 4 animals-10-00337-t004:** Analysis of the amino acid composition of each component.

Amino Acid	Fr.A2	Fr.B1
	Molar Percentage (%)	Molar Percentage (%)
Arg	10.30 ± 0.30	8.14 ± 0.30
His	9.59 ± 0.40	7.89 ± 0.20
Lys	8.24 ± 0.20	9.04 ± 0.30
Phe	3.50 ± 0.20	4.42 ± 0.30
Tyr	2.22 ± 0.20	1.37 ± 0.10
Leu	6.03 ± 0.30	9.69 ± 0.40
Ile	5.62 ± 0.40	7.37 ± 0.20
Val	4.91 ± 0.10	7.43 ± 0.20
Met	6.54 ± 0.10	9.57 ± 0.20
Pro	12.72 ± 0.30	8.43 ± 0.30
Ser	6.32 ± 0.20	5.78 ± 0.30
Thr	6.06 ± 0.10	5.93 ± 0.30
Glu	5.99 ± 0.10	4.17 ± 0.30
Gly	6.60 ± 0.10	3.09 ± 0.50
Ala	5.37 ± 0.40	7.69 ± 0.10
Hydrophobic amino acids (%)	51.29	57.69
Alkaline amino acids (%)	28.13	25.07

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
