# Peer review of "Antibacterial Activity of Trypsin-Hydrolyzed Camel and Cow Whey and Their Fractions"

_animals, 2020, doi:10.3390/ani10020337_

Round 1

Reviewer 1 Report

Dear Sir,

I thoroughly read this work again and I found that all comments were taken into account.

Best regards

Shehadeh  kaskous

Author Response

Point 1:I thoroughly read this work again and I found that all comments were taken into account.

Response 1:Special thanks to you for your good comments.

Reviewer 2 Report

There were great improvement in this revised manuscript according to comments raised by reviewer. However, the languages should be polished by native speaker.

Author Response

Point 1: There were great improvement in this revised manuscript according to comments raised by reviewer. However, the languages should be polished by native speaker.

Response 1: Special thanks to you for your good comments. We have invited the native English-speaking scientists of Elixigen Company (Huntington Beach, California) for editing our manuscript.

Reviewer 3 Report

Having reviewed the article entitled “Antibacterial activity of trypsin-hydrolyzed camel whey, cow whey, and their fractions”, I have found some shortcomings that the authors still have to improve.

Line 41-42 „and the antibacterial activity of camel milk was slightly higher than that of cow milk” – the authors analysed whey and its fractions which had been obtained after hydrolysis, so they are able to define only the antibacterial properties of this part of milk and not the whole milk. Please improve the pointed phrase.

Line 89: „Cows” –please change to „cows”.

2.1. Materials – in this paragraph the authors should add the methods of milk analysis. Fat and total protein should be measured according to the reference method. If the total protein had been determined with MilkoScan and Kiejdahl methods, the data would have been slightly different from each other. The total protein content is very important when it comes to the preparation of the samples to the electrophoresis. Therefore, these results affect the quality of the analyses performed.

Line 90, 91, 101. If it is impossible (e.g. due to editor’s orders) to use “rmp” in the description of the  centrifuge condition, round the data up or down please.

Line 134: In the place of the phrase ”dextran gel filtration chromatography (DGFC)” I suggest to use “Sephadex G-25 gel filtration column”.

Fig 1. depicting the degree of hydrolysis results is incorrect. Its’ shape does not correspond to the equation of L. Michaelis and M. Menten. During the hydrolysis it is not possible for the curve to fall down at a slightest point, no matter how  long the reaction time would be. In my opinion, the analysis of the degree of hydrolysis has to be repeated.

Fig 2. and 3. were not changed. I notice the background to have been a little bleached, but the quality of the photos still needs improvement. The SDS-PAGE analysis should be repeated.

Line 266-269: The phrase “high-quality peptides is higher molecular weight”, and the “low-quality peptides is lower  molecular weight” is not a scientific explanation.  The words “high, higher low and lower” do not exist in the scientific nomenclature. I still do not know what the authors meant. The phrases are not precise. The authors are expected to describe the obtained results in detail.

The manuscript is not adequate for publishing.

Author Response

This manuscript is a resubmission of an earlier submission. The following is a list of the peer review reports and author responses from that submission.

Round 1

Reviewer 1 Report

Dear authors,

I have marked a few comments on the text. I ask you to take These comments into account. 

I wish you success

Reviewer 2 Report

This manusript entilted Antibacterial activity of trypsin-hydrolyzed camel 2 whey, cow whey, and their fractions was carefully reviewed. Authors claimed that they investigated the antibacterial activity of camel whey proteins digested by bovin trypsin, which was similar to the results reported by the reference 12. However, its novelty is that it provided us with TEM images of antibacteria acted by peptides from camel whey. 

Author Response

Response to Reviewer 2 Comments

Point 1:Antibacterial activity of trypsin-hydrolyzed camel 2 whey, cow whey, and their fractions was carefully reviewed. Authors claimed that they investigated the antibacterial activity of camel whey proteins digested by bovin trypsin, which was similar to the results reported by the reference 12. However, its novelty is that it provided us with TEM images of antibacteria acted by peptides from camel whey.

Response 1:Special thanks to you for your good comments.

Reviewer 3 Report

The article entitled “Antibacterial activity of trypsin-hydrolyzed camel whey, cow whey, and their fractions”  by Ruixue Wang, Zhihao Han, Rimutu Ji, Yuchen Xiao, Rendalai Si, Fucheng Guo, Jing He, Le Hai , Liang Ming, Li Yi directed to Animals - Open Acces Journal presents the properties of hydrolysed milk whey originated from camel and cow.

Whey was hydrolyzed and divided into fractions, and then their antibacterial properties were determined. The experiment was properly designed, however some cardinal mistakes were found:

Line 88: The authors pointed out the amount of proteins in both kinds of milk: “Milk from cows (3.62% fat, 6.87% total protein) and camels (4.23% fat, 10.19% total protein)”. Generally, a cow’s milk contains 3-3.5% total protein and camel’s milk approximately 4-4.02%. Therefore, the question arises: How was the amount of protein determined? What was the method used? In regular analysis, the Kiejdahl method is used to measure the total protein in milk.

Milk of cows or camels does not have such a high amount of proteins. If the authors performed the rest of analyzes according to these data, the question arises whether the results obtained are correct.

Please refer to this remark and try to explain the phenomenon of a high amount of proteins reported in both kinds of milk.

Line 90, 91, 101, please explain the “strange” conditions used during centrifugation (5031 ×g, 4024.8 ×g, 2515.5 ×g,) – what was the reason to use such conditions?

Line 96: “Lyophilized CaW and CoW” – there is no information about the process of lyophilization. How it was done, what conditions were used during the process, what kind of equipment was used. Please add these details.

Line 131: “SEC” – please give the full name.

In the paragraph of “Results and discussion” it is necessary to enclose the figure with kinetics of sample hydrolysis reaction. The standard shape of curve of a hydrolysed protein has three phases: the first rising logarithmically, the second still growing, but in a  more gentle manner, and the third – the plateau.

In the presented experiment, the third part of the curve is still growing, what indicates that the hydrolysis process is not over yet and the amino groups are still being released from the proteins. This analysis should be repeated, because the presented data raise doubts as to the correctness of the analysis.

Fig.1 and 2 are both of poor quality and should be improved.

Fig. 2. It shows the SDS-PAGE of cow whey hydrolysates, but the rest of the text below the figure described camel whey hydrolysates. Please correct this mistake.

Line 2016-232. The same text was placed twice. Please remove one of them.

Table 1. In the title of the table the authors used the phrase “the antimicrobial peptides”. Analysed fractions were the cocktail of peptides. None of them was characterized, therefore it would be better to describe the table as “The antibacterial activity of the whey fractions”.

Line 255 and 256: Please describe how did you define” “low-quality peptides” and “high-quality peptides”. Since the peptides in hydrolyzate fractions  were not identified, , I do not know what the authors meant.

Table 2. Please explain what was used as a control in measurement inhibition zone diameter.

I did not find such information neither in  the“Materials and Methods” part nor in the table 2.

Table 3. Taking into consideration the size of the table, it seems that the name of the strains can be put into the table. This will facilitate reading and tracking data in the table.

Figure 4. “a, b, c: a, b, c:” – the letters were duplicated. Please correct the mistake.

Table 4. “Fr.A2” and “Fr.B1” are used in the table header. The authors used so many acronyms to describe specific fractions that the reader gets lost. I suggest you prepare a table with an accurate description of the factions so that the reader can easily figure out the information given.

Line 341: Why did the authors use trypsin to a hydrolased milk? What was the choice based on?

Line 344-345: “camel milk does not contain the sensitizing factor β-Lg, making it easier to digest and absorb” – in my opinion this conclusion is exaggerated.

Summing up, at this stage the article  is not suitable for further evaluation for Animals Journal, as it has a few very important mistakes that require in-depth reconsideration and improvement. which .
